# Lagged Association between Climate Variables and Hospital Admissions for Pneumonia in South Africa

**DOI:** 10.3390/ijerph18126191

**Published:** 2021-06-08

**Authors:** Hugo Pedder, Thandi Kapwata, Guy Howard, Rajen N. Naidoo, Zamantimande Kunene, Richard W. Morris, Angela Mathee, Caradee Y. Wright

**Affiliations:** 1Population Health Sciences, University of Bristol, Canynge Hall, Bristol BS8 2PN, UK; hugo.pedder@bristol.ac.uk (H.P.); richard.morris@bristol.ac.uk (R.W.M.); 2Environment and Health Research Unit, South African Medical Research Council, Johannesburg 2028, South Africa; thandi.kapwata@mrc.ac.za (T.K.); zama.kunene@mrc.ac.za (Z.K.); angie.mathee@mrc.ac.za (A.M.); 3Environmental Health Department, Faculty of Health Sciences, University of Johannesburg, Johannesburg 2028, South Africa; 4School of Civil, Aerospace and Mechanical Engineering, University Walk, Bristol BS8 1TR, UK; guy.howard@bristol.ac.uk; 5Discipline of Occupational and Environmental Health, University of KwaZulu-Natal, Durban 4001, South Africa; naidoon@ukzn.ac.za; 6School of Public Health, Faculty of Health Sciences, University of the Witwatersrand, Johannesburg 2000, South Africa; 7Environment and Health Research Unit, South African Medical Research Council, Pretoria 0084, South Africa; 8Department of Geography, Geoinformatics and Meteorology, University of Pretoria, Pretoria 0084, South Africa

**Keywords:** climate change, distributed non-linear lag model, environmental health, public health, respiratory disease, pneumonia, South Africa, meteorology

## Abstract

Pneumonia is a leading cause of hospitalization in South Africa. Climate change could potentially affect its incidence via changes in meteorological conditions. We investigated the delayed effects of temperature and relative humidity on pneumonia hospital admissions at two large public hospitals in Limpopo province, South Africa. Using 4062 pneumonia hospital admission records from 2007 to 2015, a time-varying distributed lag non-linear model was used to estimate temperature-lag and relative humidity-lag pneumonia relationships. Mean temperature, relative humidity and diurnal temperature range were all significantly associated with pneumonia admissions. Cumulatively across the 21-day period, higher mean daily temperature (30 °C relative to 21 °C) was most strongly associated with a decreased rate of hospital admissions (relative rate ratios (RR): 0.34, 95% confidence interval (CI): 0.14–0.82), whereas results were suggestive of lower mean daily temperature (12 °C relative to 21 °C) being associated with an increased rate of admissions (RR: 1.27, 95%CI: 0.75–2.16). Higher relative humidity (>80%) was associated with fewer hospital admissions while low relative humidity (<30%) was associated with increased admissions. A proportion of pneumonia admissions were attributable to changes in meteorological variables, and our results indicate that even small shifts in their distributions (e.g., due to climate change) could lead to substantial changes in their burden. These findings can inform a better understanding of the health implications of climate change and the burden of hospital admissions for pneumonia now and in the future.

## 1. Introduction

Pneumonia is a serious health problem especially among children under 5 years of age. In 2017, approximately 808,694 children globally died from pneumonia [1]. The most common causes of pneumonia are *Streptococcus pneumoniae* (a bacterium), *Haemophilus influenczae* type b (Hib, also a bacterial pneumonia), respiratory syncytial virus and for infants with human immunodeficiency virus (HIV), *Pneumocystis jiroveci* is the most common cause of pneumonia [2]. Pneumonia is an acute respiratory disease and is especially common among malnourished children, children with HIV, children living in crowded homes, those who are exposed to environmental tobacco smoke as well as smoke from burning biomass fuels indoors. A Global Action Plan for Pneumonia and Diarrhea by the World Health Organization and UNICEF proposes several interventions to prevent and treat pneumonia in children [3]. Prevention of pneumonia among children is through promotion of adequate nutrition (including exclusive breastfeeding for the first six months of life), good hygiene and clean air [1].

Seasonal patterns in pneumonia prevalence exist; however, they are inconsistent. A bimodal seasonal pattern (summer and winter) with substantial variability was seen among pneumonia patients in Thailand [4]. A similar pattern was seen among pneumonia patients in Kilifi, Kenya where pneumonia prevalence was found to be higher during the rainy season with a double peak in March and October [5]. A Taiwanese study found a single peak in pneumonia admissions during cold months and associated a 1 degree Celsius (°C) decrease in ambient temperature with a ~0.03% increase in monthly pneumonia admissions [6]. Given the variation in seasonality across studies of pneumonia admissions, local studies at country level are necessary to inform local prevention and health promotion campaigns.

The seasonal variations seen in pneumonia admissions have been related to climate and meteorological variables: ambient temperature, relative humidity, rainfall, and air pollution and their impact on both the host and the pathogen. Among 34,303 child (<15 years of age) pneumonia admissions in Hong Kong, temperature exhibited a U-shaped association with pneumonia with a minimum morbidity temperature of 25 °C and a long-lagged effect up to 45 days [7]. Relative humidity also showed a U-shaped association with minimum risk at 65% and lagged effect up to 45 days. In a study of Australian patients admitted for pneumonia, a time-varying distributed lag non-linear model showed that relative rate ratios (RR) associated with temperature were immediate and highest in late winter (lowest temperature of effect was 16 °C; RR: 3.10, 95% confidence interval (CI): 1.80, 5.26) [8]. For rainfall, the cumulative effect over the long range of 0–8 weeks showed two peaks for low rainfall (12 mm, RR: 2.08, 95% CI: 1.38, 3.10) and moderately high rainfall (51 mm; RR: 3.16, 95% CI: 1.11, 9.03). Again, the seasonal differences vary by region/country.

The relationship between climate change and pneumonia is complex and mediated through other factors, including socio-economic conditions, migration and exposure to extreme events. Several studies have noted that pneumonia may increase as extreme events increase, particularly in vulnerable communities with poor environmental health conditions [9,10,11,12].

In South Africa, pneumonia is the second most prevalent cause of death among children under five years of age [13]. Children who are poorly nourished, those with HIV and those who are HIV-exposed have a high risk of severe pneumonia and death compared to well-nourished and HIV-uninfected children [13]. While vaccines are one means with which to prevent severe pneumonia morbidity and mortality, so too is understanding the epidemiology of pneumonia, especially seasonal variations. Such patterns in respiratory diseases including pneumonia may be changing with a changing climate [11] and thus understanding baseline relationships between climate variables and pneumonia is essential. This is deemed important for South Africa—a country projected to experience significant changes in temperature and rainfall with climate change [14,15].

The aim of this study was to perform an exploratory analysis investigating the association between two meteorological variables, temperature and relative humidity, and pneumonia hospital admissions in Mopani District Municipality, Limpopo Province of South Africa. Given that we expect the effects of these variables to have a delayed effect on hospital admissions, we modeled their association using a framework that al-lows for investigation of this “lagged” effect, namely distributed lag non-linear models (DLNM) [16].

## 2. Materials and Methods

### 2.1. Dataset

The infectious Disease Early Warning System (iDEWS) dataset contains hospital admissions from two different hospitals in the Mopani district in South Africa (Figure 1) and has previously been used to investigate climatic factors relating to diarrhea hospital admissions [17]. Permission to conduct the study was granted by the Limpopo Department of Health (REF 4/2/2), the management staff of Nkhensani Hospital and Maphutha L. Malatjie Hospital. The South African Medical Research Council Research Ethics Committee approved the study protocol (EC005-3/2014). In total there are 59,665 admission records in the dataset, from calendar years spanning 2002–2017 (inclusive).

Pneumonia admissions were identified based on their recorded reason for admission at the time of admission, and therefore cases could be considered as “suspected pneumonia admissions” rather than confirmed pneumonia (based on a diagnostic test). Although microbiology and culture are performed routinely for patients admitted with pneumonia, this focuses on bacteriology to ensure treatment strategies are appropriate. Viral typing is not performed routinely. The following terms were used to identify pneumonia admissions within the dataset (* indicate wildcard characters): **monia**; **brpn**; **bpn**; **pcp**; **bpn**; and NOT **chem**; **pneumothorax**; **asp** (in order to exclude aspiration pneumonia). No International Classification of Diseases (ICD-10) codes were available in the admissions records.

Individual admissions were aggregated to calculate daily admission counts. Days for which data were not available in the dataset were assumed to have zero admissions. However, for the period of 1 January 2006 to 10 October 2007 no records are available, as hospital admission books could not be found. We therefore restrict our analyses to dates after 10 October 2007. Of patient admissions after this date, 9.8% were missing data on age, and 21.2% were missing data on sex; 69.0% of admissions for which age was reported were younger than 18 years, although this is likely to reflect bias towards reporting age in younger patients rather than being an accurate representation of the age distribution in the dataset.

For some cases, data were missing on the exact date of admission. As this was essential for inclusion of admissions in the time series, we estimated date of admission by subtracting the days spent in hospital from the date of discharge. If days spent in hospital were not reported, we imputed this using multiple imputation. Variables used to predict missing days spent in hospital were day of the year, day of the week, week month, year, season, average daily temperature, minimum daily temperature, maximum daily temperature, average daily relative humidity, minimum daily relative humidity, maximum daily relative humidity. Imputation was performed over 20 datasets using predictive mean matching and model results were pooled using Rubin’s rules to account for uncertainty in the imputation procedure [18].

Meteorological data were obtained from Thohoyandou where a weather station was located approximately 40 km outside of the Mopani district. Data on mean daily temperature (°C) and relative humidity (%) were directly extracted from the dataset. Daily temperature range (DTR) (°C) was calculated as the maximum minus the minimum temperature for each day. In a previous study [19], temperatures measured at the Thohoyandou weather station and in dwellings in Giyani (the location of the hospital) were well correlated (R = 0.98, *p* < 0.0001), suggesting that meteorological conditions did not vary substantially between the station and the communities under study.

### 2.2. Statistical Methodology

We examined several published studies using different DLNMs and time series models relating lagged meteorological data to pneumonia hospital admissions [20,21,22]. However, models used in these studies were difficult to replicate as insufficient information was given regarding the structure of the model and degrees of freedom/knots used for modelling different non-linear functions. Therefore, we investigated relationships with some of the key variables in these published models and used information that was available in them (e.g., regarding lag duration) for developing our own models on the iDEWS dataset.

DLNMs can be used to simultaneously model non-linear exposure-response associations and delayed effects, and as such they are particularly useful in environmental epidemiology [16]. DLNMs were fitted to the hospital admission counts using generalized linear models with a zero-inflated negative binomial likelihood that allowed for overdispersion and a higher proportion of days with zero admissions than would be expected from a typical Poisson distribution. Natural cubic spline basis functions with different numbers of knots were investigated for the exposure response associations of different meteorological predictors and their lagged effects on the rate (negative binomial) component of the model. Model selection in DLNMs is challenging due to the bi-dimensional nature of simultaneously modelling both exposure and lags [23]. If a meteorological factor was found to have a significant association (*p* < 0.05) with hospital admissions at any time across the lag periods we investigated (21 days) it was included in the model. The number of knots was selected by examining model residuals plotted against each predictor and investigating results of sensitivity analyses (see Appendix A). The fit of the final model was confirmed by examination of residuals and autocorrelation (see Appendix A).

In our final model, the rate (negative binomial) component incorporated associations with mean daily temperature and relative humidity using natural cubic splines with a lagged response over 21 days, and daily temperature range using a natural cubic spline with no lagged response. Day of the week was included as a categorical variable, a periodic seasonal effect was modelled using a Fourier series with a wavelength of 6 months, and long-term trend in admissions was modelled using a natural cubic spline. The zero-inflated (binomial) component included day of the week as a categorical variable, and we added linear terms for the number of previous pneumonia admissions in the preceding 3 days. These additional terms help to account for the feature of pneumonia as an infectious disease [24] and reduce the serial autocorrelation in the residual deviances of the model [25]. A more detailed definition of the model is given in Appendix B.

Models were fitted in R (version 4.0.2) [26] using the package *dlnm* (version 2.4.2) [23]. Analysis code is available at: https://github.com/hugaped/iDEWS_Pneumonia (accessed 19 May 2021). Results are plotted as predicted rate ratios (RR) at different exposure and lag values. Predictions were centered at median values for mean daily temperature (21 °C) and relative humidity (67%), and for DTR at the optimal lowest value in the dataset (1.3 °C) meaning that at these values, RR will be equal to 1.

The attributable risk fraction (AF) was calculated using a backward perspective as the proportion of pneumonia admissions in the dataset that were attributable to the lagged effects of each of the meteorological variables [27]. The same values were used for centering as for predictions (see above), meaning that the AF represented the proportion of admissions due to variations from these central values. A forward perspective was also taken to examine the impacts of potential shifts in the distribution meteorological variables that might arise due to long-term climate changes. Empirical 95%CIs were calculated using Monte Carlo simulation over 1000 repetitions [27].

## 3. Results

There were 4048 admissions for pneumonia between 10 October 2007 and 31 December 2015. Of these, 34.3% did not report date of admission, but date of admission could be calculated from date of discharge and imputed days spent in hospital. 4.5% of cases did not report a date of admission or discharge, so these were excluded from the analysis.

Figure 2 shows the pattern of daily pneumonia hospital admission counts and meteorological variables over time from a single example of an imputed dataset. There was a clear seasonal trend for meteorological variables, though this was less visually apparent in the daily hospital admission counts.

Model estimates in Table A1 (see Appendix C) showed a significant seasonal component affecting the odds of whether there will be pneumonia hospital admissions on a given day, and a clear association between day of the week and rate of admissions with the largest rate occurring on Monday (RR vs. Sunday: 1.20; 95%CI: 1.03, 1.39) and the lowest on the Sunday (the reference group). Hospital admissions also appeared to have increased in more recent years (spline coefficient 5 RR: 1.53; 95%CI: 1.25, 1.88). Mean temperature, relative humidity, and temperature range were all significantly associated with pneumonia hospital admissions. The deviance of the model was still very high after accounting for these variables, indicating that meteorological and chronological variables did not explain much of the variability in hospital admissions (Appendix A).

Cumulatively across the whole lag period, lower mean daily temperature (12 °C relative to 21 °C) was associated with an increased rate of pneumonia hospital admissions (RR: 1.27, 95%CI: 0.75–2.16), and higher mean daily temperature (30 °C relative to 21 °C) was associated with a decreased rate of admissions (RR: 0.34, 95%CI: 0.14–0.82; Figure 3). Lower mean daily temperature was most strongly associated with an increased rate of pneumonia hospital admissions after 7–14 days lag, whereas higher mean daily temperature was most strongly associated with a decreased rate of hospital admissions after 0–10 days lag (Figure 4 and Appendix A). The lagged effect of mean daily temperature appeared to disappear after 17 days. We also investigated the effects of using minimum and maximum daily temperature as independent predictors in the model in place of mean daily temperature (see Appendix A).

For relative humidity, the cumulative association showed that higher relative humidity (>80%) was associated with fewer hospital admissions for pneumonia, and lower relative humidity (<30%) was associated with increased admissions over the whole 21-day lag period (Figure 5). The strongest associations with pneumonia hospital admissions were at 0–5 days lag, at which high humidity was associated with decreased admissions, and at 15–21 days lag, at which low humidity was associated with increased admissions (Figure 6 and Appendix A).

Predictions for DTR suggested that higher variations (>21 °C) between maximum and minimum daily temperature were associated with increased hospital admissions (Figure 7). There was no evidence for a lagged association with admissions for this variable, meaning that this represented the association with DTR and admissions on a given day (i.e., after 0 days lag).

Other meteorological variables (total rainfall, wind speed and wind direction) were investigated but were not found to improve the model fit or further explain the variability of daily hospital admissions for pneumonia.

The proportion of pneumonia hospital admissions attributable to mean daily temperature, relative humidity, and DTR across the full lag periods are shown in Table 1. Given that most cases are accrued at more extreme exposure values, a long-term shift in the distribution of any of these exposures could lead to a substantial increase/decrease in admissions.

## 4. Discussion

This study reports the lagged association of two meteorological variables, temperature, and relative humidity, on hospital admissions for pneumonia symptoms in South Africa. We found an association between pneumonia and mean daily temperature, relative humidity, and daily temperature range (DTR), in addition to the presence of chronological effects (e.g., seasonal).

Mean daily temperature was associated with pneumonia hospital admissions after 0–14 days, with higher temperatures associated with a lower rate of admissions, and lower temperatures associated with a higher rate of admissions. This finding is supported by the established global seasonality of the influenza virus. Laboratory based studies have postulated possible explanations for these findings [28,29,30]. Lower temperatures increase transmissibility of the pneumonia-causing influenza virus, and in controlled laboratory conditions, a significantly increased transmissibility of the influenza virus was seen at 5 °C as compared to higher temperatures. At these temperatures, there was also a substantially higher shedding of the virus by the laboratory animals [28]. This could explain associations with increased hospital admissions after longer lags.

These results are largely in agreement with previous studies, though the slightly different lags of higher and lower temperatures we found here have not been reported before. Liu et al. [31] also identified an association between low mean daily temperature and increased pneumonia admissions over a slightly shorter lag period (2–5 days) than in our model. In contrast to our results, they found higher mean temperatures associated with increased admissions. However, findings from Sohn et al. [22] corroborate our results, showing that increased mean daily temperature was associated with lower rates of admissions with 0–7 days lagged effects.

We identified an association between relative humidity and hospital admissions at both shorter (0–5 days) and longer (15–21 days) lags. The association at shorter lags would suggest (if causal) that high humidity might reduce the propensity for individuals with pneumonia to go to hospital, either by decreasing symptom severity or perhaps by decreasing the likelihood of travelling to the hospital [32]. Low humidity at 15–21 days prior was associated with increased hospital admissions. Viral pneumonia infectivity has been shown to be higher at lower relative humidity as droplet aerosols can remain airborne for longer [32]. This increases the transmission of pneumonia-causing organisms, such as the influenza virus [28], which could explain the delay between humidity and hospitalization found in our model. In Mopani, low humidity is also associated with formation of dust particulates, which has recently been shown to be another route of viral influenza transmission [33].

There is debate in the literature over the effects of humidity on pneumonia transmission and hospital admissions. Some authors suggest that after controlling for other meteorological variables there is no association with relative humidity and pneumonia admissions [22,34]. Others suggest either that higher [31] or lower [35] relative humidity is associated with increased rates of admissions. Given the heterogeneity in these findings, it is possible that the effects of humidity are mediated by local factors that differ between these studies (e.g., dusty environments, population demographics).

Relative humidity has declined over the study period, which is most likely due to increasing temperatures—mean annual temperatures have risen by at least 1.5 times the global average [14]. Due to the relationship between relative humidity and temperature, for the same moisture content in the air, an increase in temperature will lead to a reduction in relative humidity. The relationship between the two is non-linear, hence why the trend for relative humidity is greater than for temperature (Figure 2). For this reason, it is important to model temperature and relative humidity simultaneously within the same model.

Our results showed that greater differences between maximum and minimum temperature increased hospital admissions for pneumonia symptoms on any given day. Large changes in temperature could exacerbate symptoms, making people more likely to go to hospital for treatment [35], and this association has been reported previously [20,22]. However, this association was only noticeable in our analysis when DTR was greater than 21 °C, which only happened 12 days each year on average across the whole dataset.

We also calculated the attributable risk of pneumonia admission for each meteorological variable, which indicated that a reasonable proportion of pneumonia admissions (either increased or decreased) were attributable to changes in mean daily temperature, relative humidity, and DTR. Given that the strongest associations between these variables were often at exposure ranges at more extreme ends of the distributions (i.e., they occurred only on a smaller number of days), the AF would change considerably if the distributions shifted even by a small amount. For example, if the distribution of mean daily temperature increased by 2 °C in line with projections for 2050 from the Intergovernmental Panel on Climate Change (IPCC) scenario RCP4.5 [14], the number of days per year for which mean daily temperature exceeds 26 °C would increase from 34 to 92, and this would have a correspondingly large effect on AF (Table 1).

Projections for relative humidity and DTR are more varied, so it is difficult to predict exactly how the distributions may shift due to climate change but, as with mean daily temperature, small changes could have a considerable impact on the burden of pneumonia hospital admissions. Increases in temperature would be likely to lead to lower relative humidity thus off-setting to some extent the benefit of increased temperature on reducing pneumonia hospitalization. However, given that changes in precipitation and relative humidity are projected with much less confidence, it remains highly uncertain exactly how future climate changes will impact on pneumonia hospitalizations.

### Limitations

While the findings were robust to the various sensitivity analyses we conducted, there were several factors that limited the conclusions that could be drawn from the data. Although we identified associations between several meteorological variables and hospital admissions, the model deviance was high, in part because daily hospital admissions were extremely variable. The factors we included in our model only explained a small proportion of the variance. Other patient-level variables may be much better predictors of hospital admissions. However, in many hospitals in South Africa, recording of patient demographics is very limited and is often performed at the discretion of healthcare staff rather than as part of a standardized process of data collection. For characteristics that were recorded (e.g., age, gender) they contained a high proportion of missing values, and we did not feel it was appropriate to assume that they were ‘Missing at Random’ nor were there sufficient predictive data to allow for imputation. One approach to reduce the variability in admissions could be to model weekly rather than daily counts. However, because the lagged meteorological effects are likely to occur over a period of days rather than weeks, modelling daily admissions allows for greater sensitivity across lagged associations.

Another limitation of the dataset was that pneumonia was not confirmed either radiologically or microbiologically but based on the admitting physician’s clinical assessment contained in the hospital admission records. There may also be selection bias as physicians may have different criteria for admitting patients, based on clinical severity, type of infection and healthcare access, choosing in some instances to treat on an outpatient basis. As the hospitals were district hospitals, referral bias is likely to have been limited, as all patients are required to be first seen at the district facility prior to referral to a regional facility.

Other factors influencing the incidence and severity of pneumonia, including socio-economic status and air pollution are not accounted for in our model, nor how they may interact with meteorological variables and climate change. Furthermore, household air pollution in particular may be an important factor here, as colder weather may lead to increased burning of fuels for household heating. We could not source reliable ambient or household air quality data that would permit this investigation, but this should be considered in future studies.

## 5. Conclusions

A proportion of pneumonia admissions were attributable to changes in meteorological variables where even small shifts in their distributions could lead to substantial changes in their burden. Understanding the impact of meteorological variables on the burden of pneumonia illness and similar conditions in the healthcare service is of interest, particularly given projected changes in climate over the next century due to global warming. Our study provides good evidence to address this issue and to consider which factors are most likely to drive changes in hospital admissions for these conditions. The findings of this study can inform a better understanding of the health implications associated with pneumonia in South Africa to support decision-making in healthcare and establish a strategy for prevention and control of the disease. The findings will also support greater efforts to understand the health implications of climate change in the country.

## Figures and Tables

**Figure 1 ijerph-18-06191-f001:**
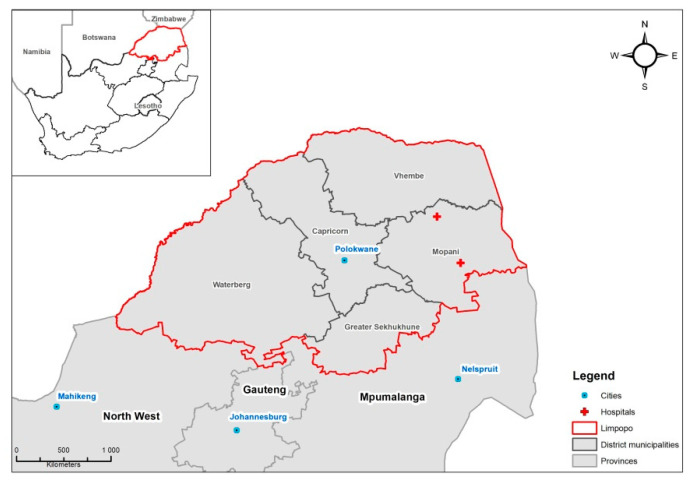
Location of the two hospitals and meteorological stations in the study site in Limpopo Province, South Africa.

**Figure 2 ijerph-18-06191-f002:**
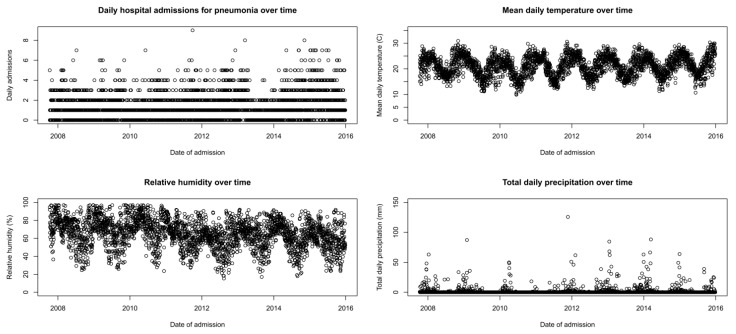
Daily pneumonia hospital admissions and meteorological variables by date.

**Figure 3 ijerph-18-06191-f003:**
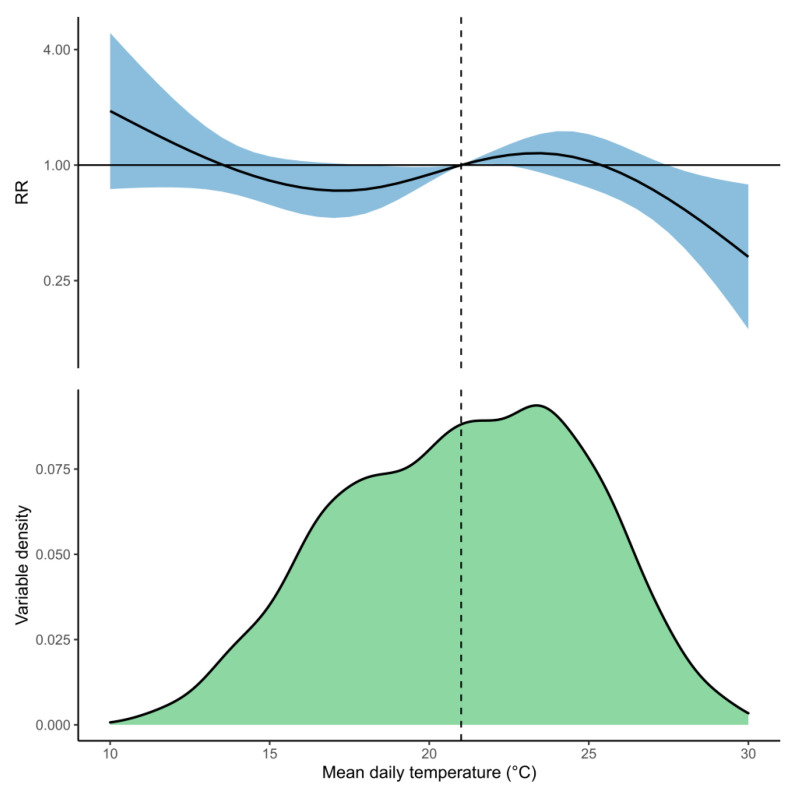
The top panel shows the cumulative association between mean daily temperature and pneumonia hospital admissions across a 21-day lag period. The solid curve is the predicted relative rates (RR), and the shaded region is the 95% confidence interval (CI). The lower panel shows the distribution of mean daily temperature within the infectious Disease Early Warning System (iDEWS) dataset. The median mean daily temperature (21 °C) is indicated by the dashed vertical line and is used as the reference value against which the RR for other temperatures is compared.

**Figure 4 ijerph-18-06191-f004:**
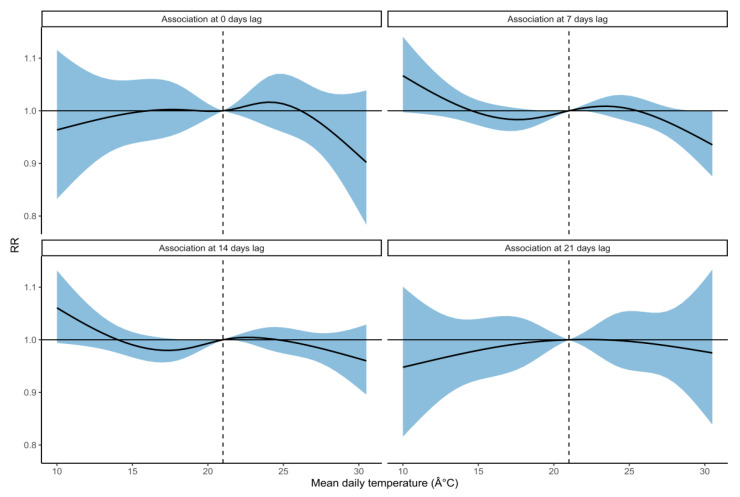
The association between mean daily temperature and pneumonia hospital admissions at different days of lag. The solid curves are the predicted relative rates (RR), and the shaded regions are the 95% CI. The median mean daily temperature (21 °C) is indicated by the dashed vertical line and is used as the reference value against which the RR for other temperatures is compared.

**Figure 5 ijerph-18-06191-f005:**
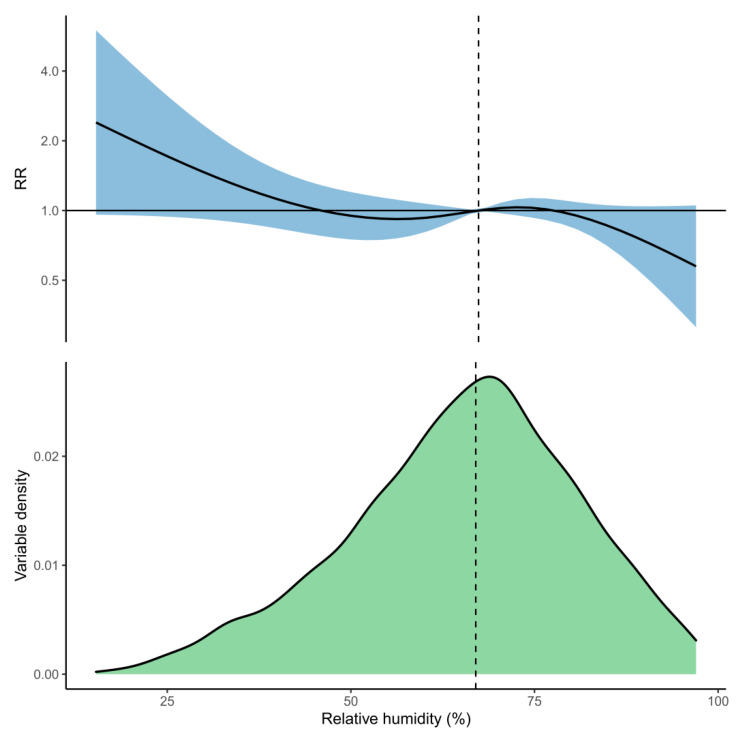
The top panel shows the cumulative association between relative humidity and pneumonia hospital admissions across a 21-day lag period. The solid curve is the predicted relative rates (RR), and the shaded region is the 95% CI. The lower panel shows the distribution of relative humidity within the iDEWS dataset. The median relative humidity (67%) is indicated by the dashed vertical line and is used as the reference value against which the RR for other relative humidity values is compared.

**Figure 6 ijerph-18-06191-f006:**
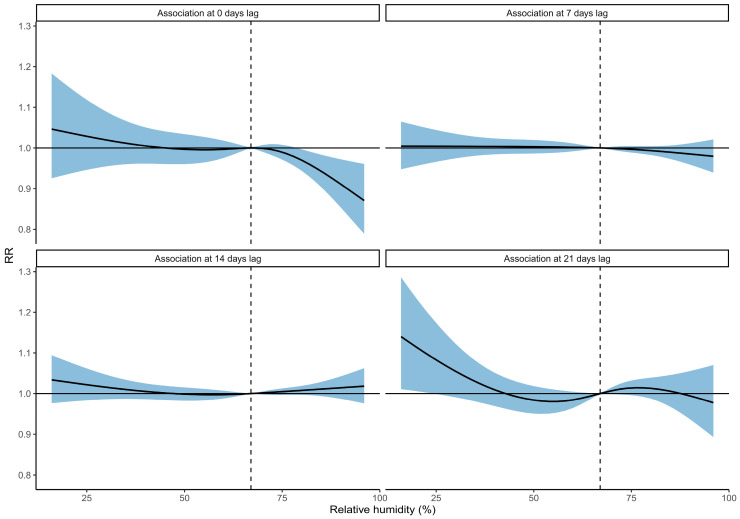
The association between relative humidity and pneumonia hospital admissions at different days of lag. The solid curves are the predicted relative rates (RR), and the shaded regions are the 95%CI. The median relative humidity (67%) is indicated by the dashed vertical line and is used as the reference value against which the RR for other relative humidity values is compared.

**Figure 7 ijerph-18-06191-f007:**
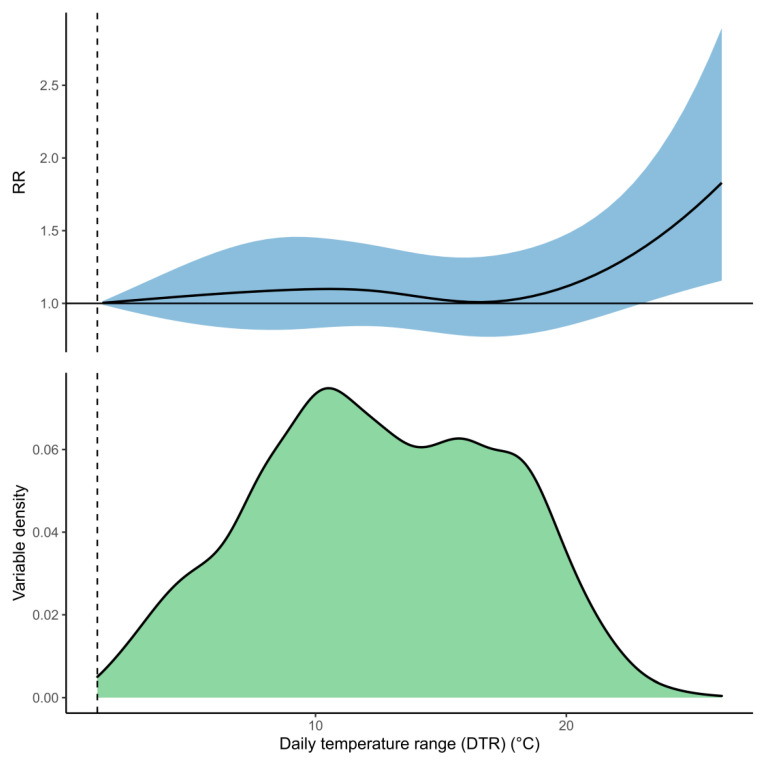
The top panel shows the association between daily temperature range (DTR) and pneumonia hospital admissions. No lagged association was modelled for this exposure, so this represents the association on any given day (i.e., zero days lag). The solid curve is the predicted relative rates (RR), and the shaded region is the 95% CI. The lower panel shows the distribution of DTR within the iDEWS dataset. The optimal lowest DTR in the dataset (1.3 °C) is indicated by the dashed vertical line and is used as the reference value against which the RR for other DTR values is compared.

**Table 1 ijerph-18-06191-t001:** Proportion of pneumonia hospital admissions attributable to changes in meteorological exposures (AF = attributable fraction), shown as the total AF, the AF due to lower values of the exposures, and the AF due to higher values of the exposures. ± changes in meteorological exposures indicate how the AF would change if the exposure distribution was shifted positively or negatively by the specified amount.

Meteorological Exposure	Total (95%CI)	Low ^a^ (95%CI)	High ^b^ (95%CI)
Mean daily temperature	−8.4% (−25%, 6.6%)	0.3% (−1%, 1.5%)	−3.4% (−8.2%, 0.3%)
+2 °C	−22.8% (−56.7%, −1.2%)	0% (−0.2%, 0.1%)	−19.2% (−54.9%, −1.4%)
−2 °C	−9.3% (−27.2%, 4%)	1.5% (−3%, 4.4%)	−0.5% (−1.4%, 0.1%)
Relative humidity	−2.1% (−12.9%, 7.7%)	2.2% (−0.8%, 4.1%)	−4.4% (−11.6%, 0.6%)
+5%	−9.3% (−29.7%, 4.6%)	1.2% (−0.5%, 2.4%)	−9.5% (−25.4%, 1.1%)
−5%	−0.6% (−13.2%, 9.4%)	3.6% (−0.6%, 6.5%)	−1.8% (−4.8%, 0.3%)
DTR	6.7% (−19.4%, 26%)	-	1.3% (−0.6%, 2.6%)
+2 °C	7.9% (−17%, 29.2%)	-	3.8% (−0.9%, 7%)
−2 °C	5.3% (−20.4%, 24.7%)	-	0.3% (−0.2%, 0.7%)

^a^ The AF for mean daily temperatures <14 °C or relative humidity <40%. ^b^ The AF for mean daily temperatures >26 °C, relative humidity >80% or DTR >20 °C.

## Data Availability

The data presented in this study are available on request from the corresponding author.

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
