# Peer review of "Lagged Association between Climate Variables and Hospital Admissions for Pneumonia in South Africa"

_ijerph, 2021, doi:10.3390/ijerph18126191_

Round 1

Reviewer 1 Report

Thank you for taking account of my previous queries. I am satisfied.

Author Response

Thank you for your constructive comments and for helping us improve the manuscript.

All the best.

Reviewer 2 Report

Thank you for your efforts in revising the manuscript and re-doing the analyses. 

Author Response

Thank you for your constructive comments and for helping us improve the manuscript.

All the best.

This manuscript is a resubmission of an earlier submission. The following is a list of the peer review reports and author responses from that submission.

Round 1

Reviewer 1 Report

  1. While it is understandable that "we do not have complete information on patient age" (l  352-353), many readers would assume that most patients would have had their age recorded, especially if children. An explanation is provided (l 386-387) but what proportion is missing? Much of the Introduction discusses pneumonia in children; what proportion of this study population (approximately) is children? Was gender recorded and did this have any influence on the results? What about other relevant variables e.g. HIV?
  2. Hospital admission for pneumonia seems to have increased in more recent years (Fig 2, l 208-209), why? As the diagnosis was made on the admitting physician's clinical assessment, is it possible that standards/criteria for this assessment have changed during the significant duration of records (2007-2017)?
  3. In addition to the explanations forwarded in the Discussion for the association between ambient temperature and pneumonia, it is also known that pneumonia is associated with air pollution and burning biomass fuels. Could this be a confounding factor i.e. more fuel is burnt for heating on cold days, leading to greater air pollution. Is this worth including as a co-variable in the Discussion? The final paragraph in the Discussion mentions a priority for future studies to include air quality (assumed, I think, as an independent variable) but would it be possible to measure air pollution inside as well as outside, as burning biomass might lead to local reduction in domestic air quality? 

Author Response

We thank the reviewers for giving their time and for their constructive comments, which we hope we have addressed below.

Reviewer comment: While it is understandable that "we do not have complete information on patient age" (l 352-353), many readers would assume that most patients would have had their age recorded, especially if children. An explanation is provided (l 386-387) but what proportion is missing? Much of the Introduction discusses pneumonia in children; what proportion of this study population (approximately) is children? Was gender recorded and did this have any influence on the results? What about other relevant variables e.g. HIV?

Response: We agree that this is important information, but in many hospitals in South Africa recording of patient demographics is very limited and is often at the discretion of the healthcare staff rather than as part of a standardized process. However, we have added details on the proportion missing on lines 126-130: “Of patient admissions after this date, 37.3% were missing data on age, and 45.2% were missing data on sex. 69.0% of admissions for which age was reported were younger than 18 years, though this is likely to reflect bias towards reporting age in younger pa-tients rather than being an accurate representation of the age distribution in the dataset.”

We also have provided a more detailed explanation of this in the Limitations on lines 383-388:

“However, in many hospitals in South Africa, recording of patient demographics is very limited and is often performed at the discretion of healthcare staff rather than as part of a standardized process of data collection. For characteristics that were recorded (e.g. age, gender) they contained a high proportion of missing values, and we did not feel it was appropriate to assume that they were ‘Missing at Random’ nor that there was sufficient predictive data to try to impute them.”

Whilst we acknowledge that these are potentially important variables for prediction and improving model fit, we do not expect them to be confounders as they are unlikely to be related to meteorological variables.

Reviewer comment: Hospital admission for pneumonia seems to have increased in more recent years (Fig 2, l 208-209), why? As the diagnosis was made on the admitting physician's clinical assessment, is it possible that standards/criteria for this assessment have changed during the significant duration of records (2007-2017)?

Response: Thanks for your comment. We have admitted to this being a limitation on lines 395-400: “There may also be selection bias as physicians may have different criteria for admitting patients, based on clinical severity, type of infection and healthcare access, choosing in some instances to treat on an outpatient basis”. However, these clinical approaches have not varied substantially over time. The increase in pneumonia admissions follows a similar trend to the increase in total hospital admissions in these hospitals and may therefore be a reflection of better access to healthcare more generally in recent years rather than changes in pneumonia-specific factors.

Reviewer comment: In addition to the explanations forwarded in the Discussion for the association between ambient temperature and pneumonia, it is also known that pneumonia is associated with air pollution and burning biomass fuels. Could this be a confounding factor i.e. more fuel is burnt for heating on cold days, leading to greater air pollution. Is this worth including as a co-variable in the Discussion? The final paragraph in the Discussion mentions a priority for future studies to include air quality (assumed, I think, as an independent variable) but would it be possible to measure air pollution inside as well as outside, as burning biomass might lead to local reduction in domestic air quality? 

Response: Thanks for the suggestion. Unfortunately, we do not have information regarding a patient’s main fuel source at home and collecting and analysing this is outside the scope of this manuscript. However, we have added a sentence acknowledging that household air pollution in particular may be an important consideration when investigating temperature on lines 417-421:

“Furthermore, household air pollution in particular may be an important factor here, as colder weather may lead to increased burning of fuels for household heating. We could not source reliable ambient or household air quality data that would permit this investigation, but this should be considered in future studies.”

Reviewer 2 Report

This study examined the association between environmental factors like temp and humidity and the number of pneumonia cases requiring hospitalization in South Africa over nine years. The analysis method is well described thought out, and necessary analyses as sensitivity analyses are well performed. Overall, the text is redundant and biased toward the analysis methods, so it would be better to significantly shorten the manuscript and rewrite it to focus on the main points.

Here is my major concern. It is important to understand that the study did not examine the association between the increase (or decrease) in the total number of pneumonia cases, but rather the relationship between the number of cases requiring hospitalization. Some seasonal viruses are known to be prevalent in a particular season like the winter season but do not cause serious diseases such as pneumonia. For instance, the RS virus or influenza which is known as the winter virus causes serious respiratory infection than a seasonal virus in the summer like the adenovirus. This point is not made clear in this study, and this point needs to be clearly stated in the text as well. Again, although not mentioned in the text, possibly the most important intermediate factor, the influence of “viruses”, was not taken into account. Therefore, it may be difficult to quantitatively state the impact of the influence of environmental factors in this study.

Here are some minor comments.

PP3 Line 120

The following terms were used to identify pneumonia admissions within the dataset (* 116 indicate wildcard characters): *aspiration pneumonia*,; *monia*; *brpn*; *bpn*; *pcp*; *bpn*; and *pneu* NOT *asp*. No ICD-10 codes were available in the admissions records.

Does this mean pneumonia caused by “any etiologies” was included? What is the assumption behind why “aspiration pneumonia” was included in this study?

Given Figure2, it seems like humidity has been declining over the study years steadily. I am wondering why and how this was considered in the analyses?

Author Response

We thank the reviewers for giving their time and for their constructive comments, which we hope we have addressed below.

Reviewer comment: This study examined the association between environmental factors like temp and humidity and the number of pneumonia cases requiring hospitalization in South Africa over nine years. The analysis method is well described thought out, and necessary analyses as sensitivity analyses are well performed. Overall, the text is redundant and biased toward the analysis methods, so it would be better to significantly shorten the manuscript and rewrite it to focus on the main points.

Response: Thank you for your review. We acknowledge that our appraisal of the literature is not based on any systematic review, but it is in light of our findings that we discuss potential mechanisms for the associations we have identified. We have shortened parts of the discussion to focus on the main points which should make the manuscript easier to read.

Reviewer comment: Here is my major concern. It is important to understand that the study did not examine the association between the increase (or decrease) in the total number of pneumonia cases, but rather the relationship between the number of cases requiring hospitalization.

Response: Yes, you are correct in pointing highlighting this. We deliberately chose “pneumonia hospitalisation” as an outcome because this reflects a more unambiguous diagnosis and indicates severity and burden of the disease on the healthcare system. To have selected a more generic outcome e.g. “pneumonia” would have certainly increased the misclassification, as this would have largely been on the basis of symptoms and clinical signs – many of which (cough, tight chest, wheeze) could mimic other respiratory disorders.

Reviewer comment: Some seasonal viruses are known to be prevalent in a particular season like the winter season but do not cause serious diseases such as pneumonia. For instance, the RS virus or influenza which is known as the winter virus causes serious respiratory infection than a seasonal virus in the summer like the adenovirus. This point is not made clear in this study, and this point needs to be clearly stated in the text as well. Again, although not mentioned in the text, possibly the most important intermediate factor, the influence of “viruses”, was not taken into account. Therefore, it may be difficult to quantitatively state the impact of the influence of environmental factors in this study.

Response: Although microbiology and culture are done routinely for patients admitted with pneumonia, this focuses on bacteriology to ensure treatment strategies are appropriate. Viral typing is not performed routinely. We have added text to the manuscript to explain this on lines 115-118:

“Although microbiology and culture are performed routinely for patients admitted with pneumonia, this focuses on bacteriology to ensure treatment strategies are appropriate. Viral typing is not performed routinely.”

Whilst it is certainly likely that prevalent seasonal viruses may account for increases/decreases in admissions, we have adjusted for seasonal components in our analysis (lines 178-179: “a periodic seasonal effect was modelled using a Fourier series with a wavelength of 6 months, and long-term trend in admissions was modelled using a natural cubic spline”), meaning that the associations with temperature reflect pneumonia admissions that are in addition to what would be expected from seasonal effects. In combination with the modelled long-term trend, the modelling of a seasonal trend would be expected to account for any effect that might arise from a particular seasonal virus. We therefore do not believe that a viral association can provide an explanation for the lagged effect that we identified in our analysis.

Reviewer comment: Here are some minor comments.

PP3 Line 120

The following terms were used to identify pneumonia admissions within the dataset (* 116 indicate wildcard characters): *aspiration pneumonia*,; *monia*; *brpn*; *bpn*; *pcp*; *bpn*; and *pneu* NOT *asp*. No ICD-10 codes were available in the admissions records.]

Does this mean pneumonia caused by “any etiologies” was included? What is the assumption behind why “aspiration pneumonia” was included in this study?

Response: Yes, our intention was to include ALL pneumonias, independent of etiology, as etiology is not specified in the admission data. We included aspiration pneumonia, as it is possible that on presentation at hospital, assumptions as to cause of the pneumonia may have been made, and exclusion could have resulted in misclassification.

Reviewer comment: Given Figure2, it seems like humidity has been declining over the study years steadily. I am wondering why and how this was considered in the analyses?

Response: This is correct that relative humidity has been declining over the study years. This is most likely due to increasing temperatures – mean annual temperatures in South Africa have increased at around 1.5 times the global increase. Due to the relationship between relative humidity and temperature, for the same moisture content in the air, an increase in temperature will lead to a reduction in relative humidity, which is why it is important to model temperature and humidity within the same model. This is not a perfectly linear relationship, hence why the trend for relative humidity is stronger than for temperature. We have added a paragraph explaining this on lines 344-351:

“Relative humidity has declined over the study period, which is most likely due to increasing temperatures – mean annual temperatures have risen by at least 1.5 times the global average[14]. Due to the relationship between relative humidity and temperature, for the same moisture content in the air, an increase in temperature will lead to a reduction in relative humidity. The relationship between the two is non-linear, hence why the trend for relative humidity is greater than for temperature (Figure 2). For this reason, it is important to model temperature and relative humidity simultaneously within the same model.”

Accounting for this long-term decline in relative humidity specifically in the analysis is not essential because our study is investigating associations between meteorological variables and health outcomes rather than specifically the decline. The long-term impact of this on health outcomes will be accounted for by the modelling of a long-term trend in pneumonia admissions. However, the effects of the long-term decline on health outcomes would be an interesting focus for future research, as we have only explored this in a fairly limited way through the use of attributable fractions of pneumonia admissions.

Round 2

Reviewer 2 Report

Thank you for responding to the comments.

I have found that the manuscript has been improved for most of the part; however, the discussion part is still too long and some of my concerns have not been addressed properly yet.

For instance, to the question of classification using "aspiration", the authors answered that "Yes, we intended to include ALL pneumonia, independent of etiology, as etiology is not specified in the admission data. We included aspiration pneumonia, as it is possible that on presentation at hospital, assumptions, as to cause of pneumonia, may have been made, and exclusion could have resulted in misclassification." 

However, I think this procedure including aspiration pneumonia might have caused "misclassification" instead. I understand the rationale in the responses noted. But the question might be "How can we explain "aspiration pneumonia" is affected by environmental factors like temperature? Anyhow, if this study includes "any type of pneumonia", this should be addressed in the limitation section clearly. This is critically important for potential readers in healthcare professions.